# The Effect of Different Anesthetic Techniques on Proliferation, Apoptosis, and Gene Expression in Colon Cancer Cells: A Pilot In Vitro Study

Alexandru Leonard Alexa [1,2,3,*], Ancuta Jurj [4], Ciprian Tomuleasa [5,6,7], Adrian Bogdan Tigu [7], Raluca-Miorita Hategan [2] and Daniela Ionescu [1,2,3,8]

1   1st Department of Anesthesia and Intensive Care, "Iuliu Haţieganu" University of Medicine and Pharmacy, 400347 Cluj-Napoca, Romania
2   Department of Anesthesia and Intensive Care, The Regional Institute of Gastroenterology and Hepatology, "Prof. Dr. Octavian Fodor", 400394 Cluj-Napoca, Romania
3   Research Association in Anesthesia and Intensive Care (ACATI), 400394 Cluj-Napoca, Romania
4   The Research Center for Functional Genomics, Biomedicine and Translational Medicine, "Iuliu Haţieganu" University of Medicine and Pharmacy, 400347 Cluj-Napoca, Romania
5   Department of Hematology, "Iuliu Haţieganu" University of Medicine and Pharmacy, 400347 Cluj-Napoca, Romania
6   Department of Hematology, Ion Chiricuta Clinical Cancer Center, 400015 Cluj-Napoca, Romania
7   Research Center for Advanced Medicine—MedFUTURE, Department of Translational Medicine, "Iuliu Haţieganu" University of Medicine and Pharmacy, 400347 Cluj-Napoca, Romania
8   Outcome Research Consortium, Cleveland, OH 44195, USA
*   Correspondence: alexandru_reziati@yahoo.com

**Abstract:** Background: Colorectal cancer is highly common and causes high mortality rates. Treatment for colorectal cancer is multidisciplinary, but in most cases the main option remains surgery. Intriguingly, in recent years, a number of studies have shown that a patient's postoperative outcome may be influenced by certain anesthetic drugs. Our main objective was to compare the effect of propofol–total intravenous anesthesia (TIVA) with sevoflurane anesthesia and to investigate the potential role of intravenous lidocaine on colon cancer cell functions. We tested the effects of serum from colorectal cancer patients undergoing TIVA vs. sevoflurane anesthesia with or without lidocaine on HCT 116 cell lines; on proliferation, apoptosis, migration, and cell cycles; and on cancer-related gene expressions. Methods: 60 patients who were scheduled for colorectal cancer surgery were randomized into four different groups (two groups with TIVA and two groups with sevoflurane anesthesia with or without intravenous lidocaine). Blood samples were collected at the start and at the end of surgery. HCT 116 cells were exposed to the patients' serum. Results: 15 patients were included in each of the study groups. We did not find any significant difference on cell viability or apoptosis between the study groups. However, there was an increased apoptosis in propofol groups, but this result was not statistically significant. A significant increase in the expression profile of the *TP53* gene in the propofol group was registered (*p* = 0.029), while in the other study groups, no significant differences were reported. *BCL2* and *CASP3* expressions increased in the sevoflurane–lidocaine group without statistical significance. Conclusions: In our study, serum from patients receiving different anesthetic techniques did not significantly influence the apoptosis, migration, and cell cycle of HCT-116 colorectal carcinoma cells. Viability was also not significantly influenced by the anesthetic technique, except the sevoflurane–lidocaine group where it was increased. The gene expression of *TP53* was significantly increased in the propofol group, which is consistent with the results of similar in vitro studies and may be one of the mechanisms by which anesthetic agents may influence the biology of cancer cells. Further studies that investigate the effects of propofol and lidocaine in different plasma concentrations on different colon cancer cell lines and assess the impacts of these findings on the clinical outcome are much needed.

**Keywords:** colon cancer; propofol; lidocaine; tumor suppressor gene; cell culture; anesthesia

## 1. Introduction

Globally, colorectal cancer (CRC) is highly common and causes high mortality rates, with almost 1 million annual deaths attributable to cancer in population with an increasing life expectancy; thus, a clinical challenge still remains [1]. Colorectal cancer treatment is multidisciplinary depending on the cancer stage, the patients' co-morbidities, etc. However, the mainstay treatment for colorectal cancers for most patients is surgery. Recent data have shown that during the perioperative period, patient's immune system and cancer cells functions may be influenced by anesthetic drugs, some of which have shown to have an impact on cancer recurrence and metastasis [2–4], although long-term survival may be influenced by numerous clinical, demographic, and genetic factors.

A number of studies, mostly in vitro or retrospective clinical studies, have compared the two main anesthetic techniques—total intravenous anesthesia (TIVA) and inhalation anesthesia—and found that TIVA is followed by a better long-term survival, lower mortality rates, and disease-free survival when compared with inhalation anesthesia in cancer patients undergoing surgery [5–9]. A number of studies have shown that propofol has direct antiproliferative and apoptotic effects in different cancer cells, as well as indirect, diminished, or absent immunosuppressive effects when compared with inhalation agents [4,6].

On the other hand, a number of studies focusing on long-term survival after propofol–TIVA and volatile anesthesia, mostly retrospective as well, found no differences in 5-year survival rates, especially in certain types of cancer such as breast cancer [10].

Local anesthetics, in the context of regional or neuraxial anesthetic techniques, have also been investigated by a small number of studies in colorectal cancer patients under the hypothesis that the immune function and stress response during surgery is less affected compared to the exclusive use of opioids, but results are not clear in studies that compare breast and prostate cancer where local anesthetics have been associated with lower recurrence rates [11,12]. Moreover, lidocaine, an amide local anesthetic, has demonstrated anti-inflammatory effects in vitro and in vivo, and is currently under investigation in relation to cancer recurrence and metastasis in large prospective clinical trials [12,13].

Most in vitro and in vivo studies have focused on breast cancer, and have investigated the effects of anesthetic techniques on different functions of oestrogen-receptor-negative cells. It was concluded that cell proliferation and other functions were inhibited by the serum from patients that were administered with TIVA–regional anesthesia, significantly more than the serum from patients undergoing volatile–opioid anesthesia [14,15]. The apoptosis of breast cancer cells was also predominantly reduced with propofol–TIVA [16]. However, other studies found that the anesthetic choice did not seem to influence cell viability or migration in breast cancer [14,16], or even the postoperative short- and long-term outcome of patients [17]. To our knowledge, besides a number of in vitro studies which investigate the effect of anesthetic drugs (including lidocaine) on colon cancer cells [18,19], there is no study that investigates colon cancer cell functions when exposed to serum from patients with colon cancer undergoing TIVA or sevoflurane anesthesia with or without the continuous infusion of lidocaine and there is no clinical study that investigates long-term outcomes in colorectal cancer patients exposed to the perioperative intravenous infusion of lidocaine.

Thus, our study aimed to investigate the magnitude of the potential effects of serum from colorectal cancer surgical patients subject to different anesthetic techniques on the HCT 116 cell line. We chose HCT-116 as it is the most common cell line used in colon cancer cell line studies [20,21] and is useful for allowing potential correlations and comparisons with the results of our group's previous study [19]. We tested the hypothesis that different anesthetic techniques have different effects on the viability, proliferation, apoptosis, migration, and cell cycle of HCT-116 cells. With this anesthetic technique, we compared propofol–TIVA vs. sevoflurane anesthesia with or without the intravenous infusion of lidocaine. Lidocaine is the only local anesthetic whose pharmacokinetic–pharmacodynamic (Pk/PD) profile is suitable for continuous infusion (allowing a greater plasma concentration

compared to neuraxial analgesia/anesthesia) and is currently used as an adjuvant for the management of perioperative pain [22].

As a second objective of the study, we aimed to investigate the effect of patients' serum on the expression of the most common genes implicated in colorectal cancer using extracted ribonucleic acid (RNA) from cancer cells and reverse transcription quantitative real-time PCR (RT-qPCR).

## 2. Materials and Methods

### 2.1. Patient Selection and Randomization

After gaining approval from the Ethics Committee of University of Medicine and Pharmacy "Iuliu Haţieganu" Cluj-Napoca (Nr. 436 from 26 November 2018) and written informed consent, patients undergoing intended curative surgery for colorectal cancer were randomized into a prospective clinical trial (registration number: NCT04162535) [23]. Randomized selected patients also consented for a 10 mL sample of venous blood from a separate peripheral venous puncture before the start of surgery (T0) and one at the end of the procedure (T1) in order to study the influence of the anesthetic technique on cancer cell biology. Blood samples were centrifuged at $400 \times g$, for 10 min at room temperature. The resulted serum was stored in aliquots at $-80\ ^\circ$C until further investigations were carried out. Inclusion criteria included patients aged 18–80, with an ASA risk score of I–III, undergoing elective colorectal surgery. Exclusion criteria were any contraindication to the drugs used in the trial, preexisting chronic pain or medication interfering with pain, major psychiatric afflictions, convulsive disorders that require medication 2 years prior, preplanned regional analgesia and/or anesthesia (spinal, epidural), hepatic or renal dysfunction, corticosteroid-dependent asthma, autoimmune illness, and any prior antiarrhythmic treatment that may interfere with the similar effects of lidocaine. Patients were randomized, with the help of a computer-generated number table, into four study groups with 15 patients each: group S—sevoflurane, group P—TIVA, group PL—TIVA with lidocaine, and group SL—sevoflurane with lidocaine.

The pooled serum samples taken from the each of the four study groups were used on the HCT-116 cell line to determine the effects on cellular viability, apoptosis, cell cycle, and cell migration. These were assessed using the MTT assay for cellular viability, fluorescence microscopy for apoptosis evaluation, flow cytometry for the assessment of cell cycle, and the wound-healing assay for assessing cell migration.

### 2.2. Anesthetic Technique

All patients included in the study were administered a prophylactic dose of LMWH 12 h prior to surgery. Anesthesia was induced in all groups with 2–3 µg/kg of fentanyl, 1.5–2 mg/kg of propofol, and 0.5–0.6 mg/kg of atracurium–rocuronium (additional boluses were given when necessary at anesthesiologist's discretion; curarization was not objectively monitored). The maintenance of anesthesia was achieved with sevoflurane in an oxygen–air mixture (MAC) of 1–1.5, depending on BIS values, for the S and SL groups. For the P and PL groups, the anesthesia was maintained with TCI propofol, in the Schnider model, with an initial effect site concentration of 2 µg/mL, adjusted accordingly to BIS values. A perioperative intravenous 1% lidocaine infusion was administered in the SL and PL groups, respectively. At the induction of anesthesia (through a peripheral venous catheter), a bolus dose of 1.5 mg/kg lidocaine was used, and, during surgery, the infusion was maintained at 1–2 mg/kg per hour with a maximum dose of 200 mg/h; doses were decreased if bradycardia or hypotension could be attributed to lidocaine.

Intraoperative analgesia was achieved with fentanyl in increments of 1–1.5 µg/kg, at the discretion of the anesthetist, in addition to a dose of 1 g of intravenous acetaminophen and a bolus dose of 0.1–0.15 mg/kg morphine, administered 30 to 45 min prior to extubation. After surgery, all patients received 0.025–0.05 mg/kg of morphine and 1 g of acetaminophen every 6 h to maintain an NRS $\leq$ 3.

## 2.3. Cell Culture

We used the HCT-116 colon cancer cell line, purchased from ATCC (American Type Culture Collection, Manassas, VA, USA). HCT-116 cells were cultured in McCoy's medium (Gibco, cat no 22330021, Carlsbad, CA, USA) supplemented with 10% fetal bovine serum (FBS) (Gibco, cat no A5256701) and 1% penicillin–streptomycin (Gibco, cat no 15070063). Cells were maintained in the incubator at 37 °C with 95% air and 5% $CO_2$. HCT-116 is a model colon cancer line; in addition, in the laboratory, the HCT-116 line was modified to express Luciferin, so we chose the HCT-116 line as it can translate the study very easily to an in vivo model.

## 2.4. Cellular Viability

The antiproliferative effects of serum were evaluated using the 3-(4,5-dimethylthiazol-2-yl)-2,5-diphenyltetrazolium bromide (MTT, Invitrogen, cat no M6494, Carlsbad, CA, USA) assay. At the sub-confluence of cells, HCT-116 cells were washed with phosphate buffer saline (PBS) 1X (Gibco, cat no 10010002) and detached with 0.05% trypsin–EDTA (Gibco, cat no 25300054). After trypsinization, $1 \times 10^4$ cells were seeded in 96-well plates and incubated for 24 h at 37 °C with 95% air and 5% $CO_2$. Cells were treated with serum which was diluted in the medium with 10% serum concentrations and incubated for an additional 24 h. After incubation, the supernatant was removed and a 1 mg/mL MTT solution was added and incubated for 1 h, and then 100 µL of dimethyl sulfoxide (DMSO) (Sigma-Aldrich, cat no D8418, St. Louis, MO, USA) was added to dissolve the formazan crystal. The MTT absorbance was measured at 570 nm using a BioTek Synergy spectrophotometer (Winooski, VT, USA).

## 2.5. Cell Cycle Evaluation by Flow Cytometry

Flow cytometry was used to investigate if there were changes in the cell cycle phases and if certain anesthetic drugs (such as lidocaine) had the potential to inhibit the cell cycle phases or cell division [24]. At a seeding density of $1 \times 10^5$, cells were cultured in 12-well plates and incubated for 24 h at 37 °C with 95% air and 5% $CO_2$. Twenty-four hours after treatment with 10% serum, cells were detached with 0.05% trypsin–EDTA and washed with cold PBS 1X. Cells were further fixed for 45 min at 4 °C with 75% ethanol which was previously cooled at −20 °C. After incubation, the fixed cells were washed again with PBS 1X. The cell pellet was resuspended in RNase buffer (0.2% BSA salt, 40 µg/mL of Rnase A, PBS 1X—prepared in house), followed by an incubation procedure which lasted for 15 min at room temperature in the dark. After incubation, propidium iodide (PI) (Invitrogen, cat no P1304MP) was added, and the analysis was performed after 30 min of incubation at room temperature in the dark. The cell cycle was performed using a BD FACSCanto II flow cytometer and the data were analyzed with FACS Diva software (version 6.0).

## 2.6. The Assessment of Apoptosis Using Fluorescent Microscopy

Apoptotic effects were evaluated using the multi-parameter apoptosis assay kit (Cayman cat no 600330, Tallinn, Estonia) on an Olympus IX71 inverted microscope (Tokyo, Japan). Cells were cultured in 96-well plates at a seeding density of $1.2 \times 10^4$ and incubated for 24 h. Twenty-four hours later, the cells were treated with 10% serum. To evaluate apoptosis, we then double-stained the cells using the Hoechst solution (Invitrogen, cat no H3570) for the nuclei with ex/em at 361/497 nm and tetramethyl rhodamine ethyl ester (TMRE) (Invitrogen, cat no T669) for the mitochondrial membrane potential with ex/em at 560/595 nm. We used the Hoechst solution as a stain for apoptotic cells after it bonded to chromatin and produced a fluorescent signal. Thus, by using Hoechst we obtained images from a large area of the flask, allowing us to determine the percentage of the apoptotic and non-apoptotic cells using our fluorescent inverted microscope.

### 2.7. Wound-Healing Assay

The changes that occurred in cell migration were evaluated via the wound-healing assay. In brief, $2 \times 10^5$ cells per well were seeded in 24-well plates and the cells were treated with 10% serum after 24 h. Twenty-four hours following administration, the wound was performed with 20 µL tips. The wounds were visualized with the Olympus IX71 inverted microscope at 0, 24, 32, and 48 h.

### 2.8. RNA Extraction and RT-qPCR

The RNA was further used for the rtPCR analysis of gene expression. We chose to analyze the TP53 gene since this is implicated in p53 synthesis and DNA signal damage. We also proceeded to analyze the TNFa, CASP3, and BCL2 genes involved in apoptosis and the MMP7 gene involved in cell migration and invasion; the TGF beta pathway was revealed by evaluating the expression of TGF beta and SMAD4 genes [25,26]. HCT-116 cells were incubated in 6-well plates at a seeding density of $3 \times 10^5$ per well and incubated for 24 h at 37 °C to allow the cells to attach. After incubation, cells were treated with 20% pooled serum and incubated for an additional 24 h. RNA was extracted from HCT-116 cells using TriReagent (Invitrogen, cat no AM9738) according to manufacturer's instructions. The RNA concentration and quality were assessed using the Nanodrop-1000 spectrophotometer (Thermo Scientific, Waltham, MA, USA). Then, 1000 ng of the total RNA was reverse-transcribed into cDNA using the high-capacity cDNA reverse transcription kit (Applied Biosystems, cat no 4368814, Foster City, CA, USA). An evaluation of the gene expression profile was conducted using the SYBR green master mix (Applied Biosystems, cat no 4344463) and qRT-PCR was performed on the ViiATM7 System in a 10 µL reaction using 384-well plates. As an internal control, we used GAPDH and beta-ACTIN. Relative quantification was conducted using the $2^{-\Delta\Delta CT}$ method.

### 2.9. Statistical Analysis

Data analysis was performed using R 4.0.1. Continuous variables were represented as the mean $+/-$ standard deviation. Differences between two groups were assessed using Welch's *t*-test. Interactions between the two variables were assessed using the two-way ANOVA. A p-value under 0.05 was considered statistically significant. Since this was a pilot study, we did not prospectively calculate a sample size to detect changes in the cancer cell's viability or apoptosis.

## 3. Results

We used the pooled serum from 15 patients from each study group. The patients' baseline characteristics, the ASA physical status, the type of surgery, the duration of anesthesia, and the tumor classification in study groups are shown in Table 1. As shown in Table 1, demographic characteristics were similar in the study groups. Preoperative chemotherapy was performed in two patients in the S and P groups and one patient in the SL and PL groups.

### 3.1. Cell Viability

The cellular viability results in the study groups (S—sevoflurane, P—TIVA, PL—TIVA with lidocaine, and SL—sevoflurane with lidocaine) are shown in Figure 1. The only statistical difference on cell viability was observed when comparing pre- and postoperative sevoflurane + lidocaine, when postoperative serum increased the cellular viability levels ($p = 0.028$). For the other study groups, there was no statistical difference in the influence on cell viability.

**Table 1.** Patients' colorectal tumor characteristics. The data are shown as the mean (standard deviation) or n = patients' number (%).

| Characteristic | Trial Groups | Group S (n = 15) | Group P (n = 15) | Group PL (n = 15) | Group SL (n = 15) | p |
|---|---|---|---|---|---|---|
| Age (years) | | 65.67 (47–80) | 61.87 (38–80) | 66.20 (57–80) | 63.67 (48–75) | |
| Gender (n,%) | Male | 6 (40) | 7 (46.66) | 10 (66.66) | 8 (53.33) | |
| | Female | 9 (60) | 8 (53.33) | 5 (33.33) | 7 (46.66) | |
| BMI (kg/m$^2$) | | 26.98 (5.28) | 26.4 (3.14) | 29.07 (5.01) | 27.87 (4.14) | |
| ASA (n,%) | 2 | 11 (73.33) | 13 (86.66) | 7 (46.66) | 13 (86.66) | |
| | 3 | 4 (26.66) | 2 (13.33) | 8 (53.33) | 2 (13.33) | |
| Preoperative chemotherapy (n,%) | | 2 (13.33) | 2 (13.33) | 1 (6.66) | 1 (6.66) | |
| Type of surgery (n,%) | | | | | | |
| Right hemicolectomy | | 5 (33.33) | 6 (40) | 8 (53.33) | 5 (33.33) | |
| Left hemicolectomy | | 3 (20) | 1 (6.66) | 1 (6.66) | 3 (20) | |
| Sigmoid colon resection | | 3 (20) | 4 (26.66) | 3 (20) | 1 (6,66) | |
| Rectosigmoid resection | | 3 (20) | 2 (13.33) | 3 (20) | 4 (26.66) | |
| Anterior rectal resection | | 1 (6.66) | 1 (6.66) | | 2 (13.33) | |
| Surgical technique (n,%) | | | | | | |
| Laparotomy | | 12(80) | 11(73.33) | 12(80) | 13(86.66) | |
| Laparoscopy | | 3(20) | 4(26.66) | 3(20) | 2(13.33) | |
| TNM classification | | | | | | |
| Pathology stage, tumor (n,%) | tx | | | | | |
| | tis | 1 (6.66) | | | | |
| | T0 | | | | | |
| | t1 | 2 (13.33) | 2 (13.33) | 1 (6.66) | 1 (6.66) | |
| | t2 | 2 (13.33) | | 1 (6.66) | 2 (6.66) | |
| | t3 | 6 (40) | 7 (46.66) | 11 (73.33) | 3 (6.66) | |
| | t4 | 4 (26.66) | 6 (40) | 2 (13.33) | 4 (6.66) | |
| Pathology stage, nodes | Nx | | | | | |
| | n0 | 10 (66.66) | 10 (66.66) | 9 (60) | 9 (60) | |
| | n1 | 4 (26.66) | 2 (13.33) | 6 (40) | 5 (33.33) | |
| | n2 | | 3 (20) | | 1 (6.66) | |
| | n3 | 1 (6.66) | | | | |
| Pathology stage, metastasis | mx | 13 (86.66) | 15 (100) | 15 (100) | 15 (100) | |
| | m0 | 1 (6.66) | | | | |
| | m1 | 1 (6.66) | | | | |
| Length of anesthesia (min) | | 156 (±52.52) | 150.33 (±48.79) | 161.33 (±59.36) | 170 (±42.68) | >0.05 |
| Fentanyl intraoperative dose (micrograms) | | 650 (±111.80) | 633.33 (±174.91) | 660 (±113.70) | 746.66 (±118.72) | >0.05 |
| Morphine intraoperative dose (mg) | | 8.53 (±2.03) | 8.73 (±2.40) | 7.26 (±1.27) | 7.53 (±0.99) | >0.05 |
| Propofol Ce intraoperative highest dose (mcg/mL) | | | 2.88 (±0.35) | 2.58 (±0.32) | | >0.05 |
| Lidocaine mean intraoperative infusion dose (except induction dose) (mg/kg/h) | | | | 1.51 (±0.18) | 1.62 (±0.34) | >0.05 |

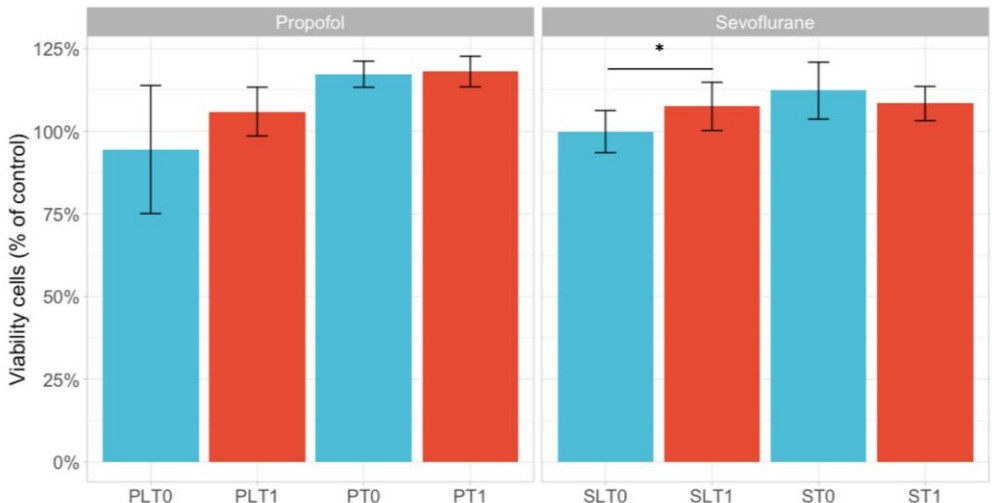

**Figure 1.** An evaluation of cellular viability levels on the HCT-116 cell line using the MTT assay of serum from patients in our study groups. The data are represented as the mean $+/-$ standard deviation and the asterisks (*) indicate significant difference at $p < 0.05$. The study groups are as follows: TIVA with lidocaine (group PL, before surgery—T0 and after surgery—T1); TIVA (group P, before surgery—T0 and after surgery—T1); sevoflurane with lidocaine (group SL, before surgery—T0 and after surgery—T1); and sevoflurane (group S, before surgery—T0 and after surgery—T1).

*3.2. Cell Apoptosis*

The results on cancer cell apoptosis when exposed to patients' serum are shown in Figure 2. As shown in Figure 2, when assessing the difference between pre- and postoperative serum apoptotic effects in each study group, we found no significant difference in any of the study groups. However, as shown in Figure 2, apoptosis was mostly expressed in the P and PL groups.

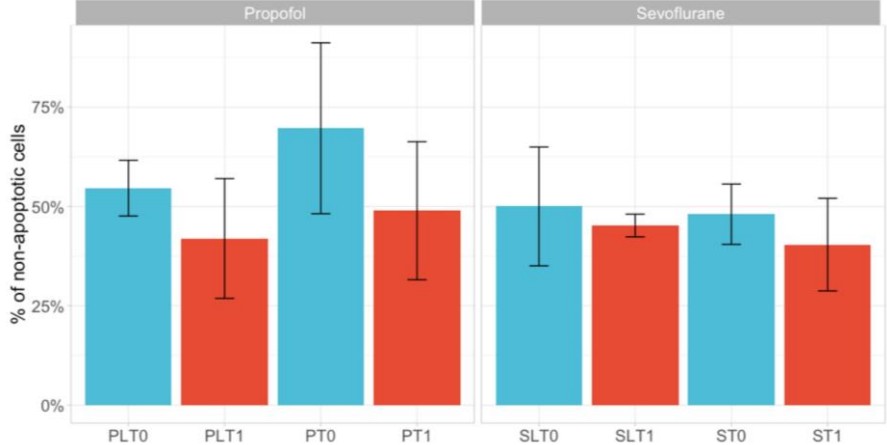

**Figure 2.** Apoptosis (fluorescence microscopy) in study groups following exposure to TIVA with lidocaine (group PL, before surgery—T0 and after surgery—T1); TIVA (group P, before surgery—T0 and after surgery—T1); sevoflurane with lidocaine (group SL, before surgery—T0 and after surgery—T1); sevoflurane (group S, before surgery—T0 and after surgery—T1) (20× magnification). The data are represented as the mean $+/-$ standard deviation.

*3.3. Cell Cycle Evaluation*

The cell cycle evaluation in the HCT-116 cell line by flow cytometry was assessed 24 h after exposure to 10% serum concentrations. There were no statistically significant differences in the cell cycle distributions between any of the groups (Figure 3).

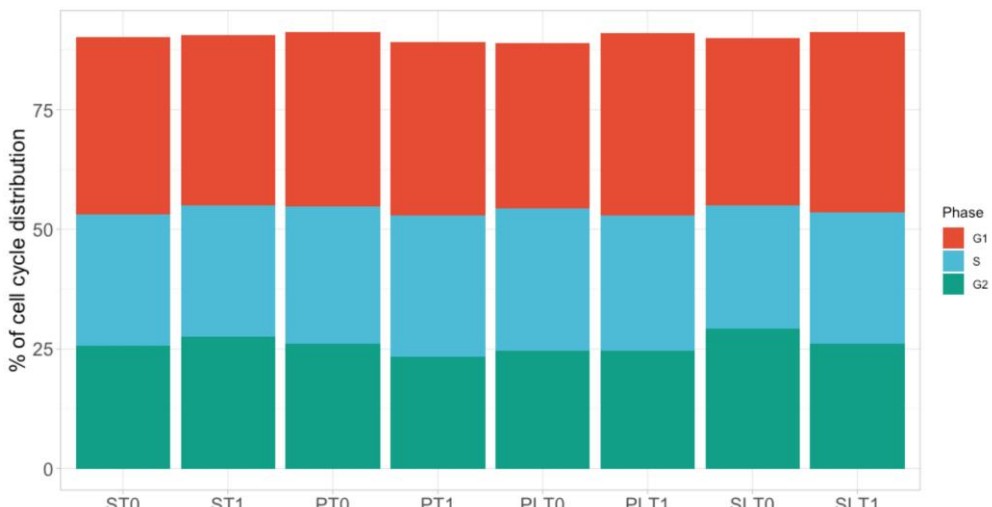

**Figure 3.** Cell cycle phase evaluation of the HCT-116 cell line in our study groups: TIVA with lidocaine (group PL, before surgery—T0 and after surgery—T1); TIVA (group P, before surgery—T0 and after surgery—T1); sevoflurane with lidocaine (group SL, before surgery—T0 and after surgery—T1); sevoflurane (group S, before surgery—T0 and after surgery—T1).

### 3.4. Wound-Healing Assay

To evaluate the effect of anesthetic intervention on the invasion of colon cancer cell lines in vitro, a wound-healing assay (control and treated cells) was performed. After 24 h, we observed that the experimental conditions did not inhibit cell invasion in the HCT-116 cell line (Figure 4).

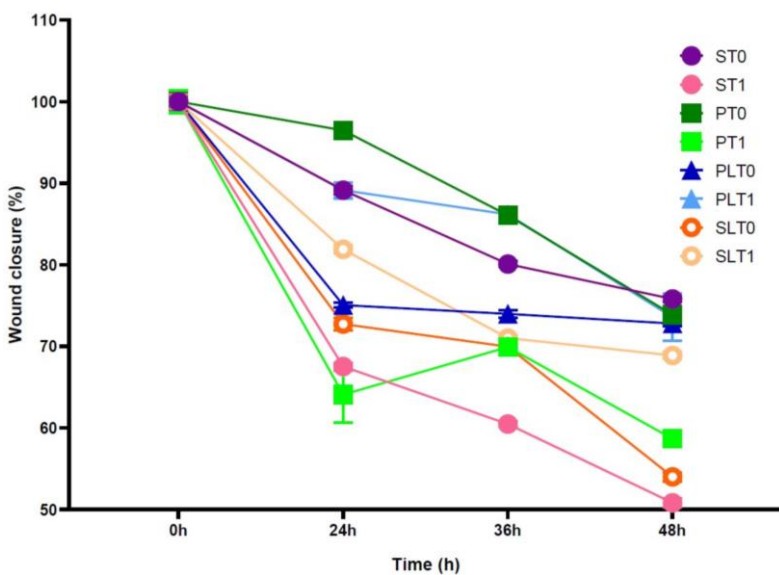

**Figure 4.** Cell migration ability was assessed via the wound-healing assay on the HCT-116 cell line. A quantitative representation of the wound distance measurement in μm is related to the time of treatment action. The data are represented as the mean +/− standard deviation (n = 3). Study groups: TIVA with lidocaine (group PL, before surgery—T0 and after surgery—T1); TIVA (group P, before surgery—T0 and after surgery—T1); sevoflurane with lidocaine (group SL, before surgery—T0 and after surgery—T1); sevoflurane (group S, before surgery—T0 and after surgery—T1).

### 3.5. RNA Extraction

Results from RNA extraction and RT-qPCR are shown in Figure 5. The expression level of several genes (*TNF-α*, *VEGFA*, *CASP3*, *BCL2*, *SMAD4*, *MMP7*, *TP53*, and *TGF-β1*) was analyzed 24 h after exposure to a solution of 20% serum.

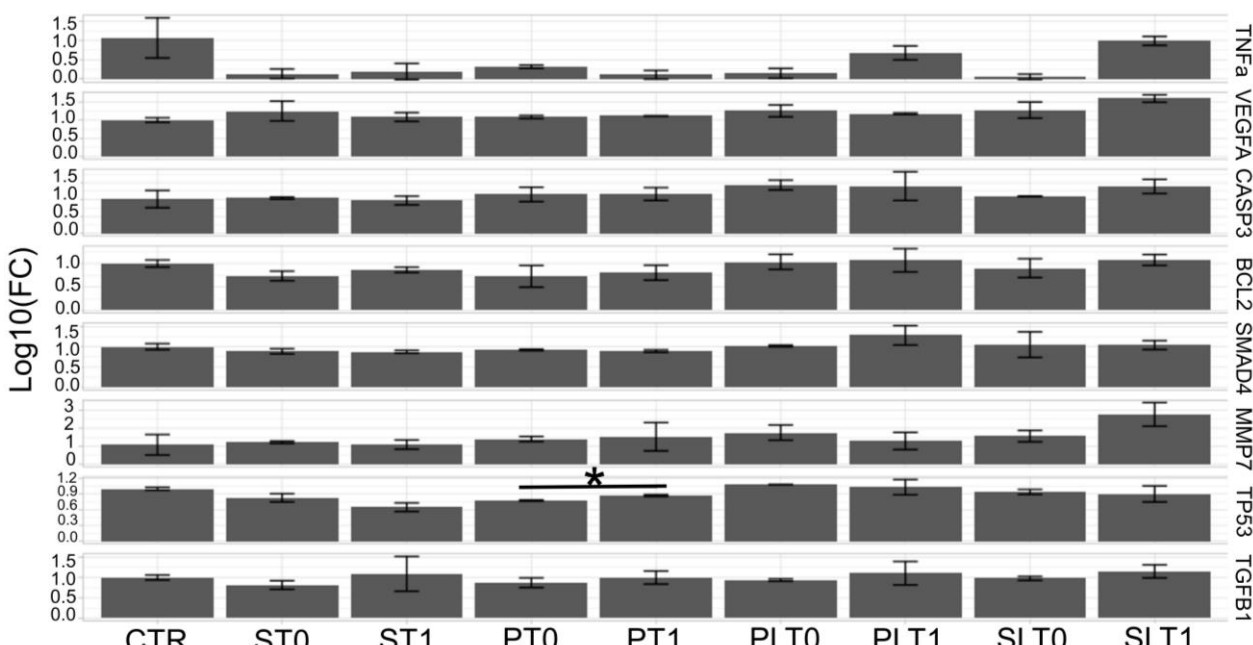

**Figure 5.** An evaluation of the gene expression levels in the HCT-116 cell line using qRT-PCR. The data are normalized to GAPDH and beta-ACTIN using the ΔΔCT method (CTR—untreated cells; group PL, before surgery—T0 and after surgery—T1; group P, before surgery—T0 and after surgery—T1; group SL, before surgery—T0 and after surgery—T1; group S, before surgery—T0 and after surgery—T1). The data are represented as mean $+/-$ standard deviation using the two-sided *t*-test. The data are represented on a logarithmic scale and the asterisks (*) indicate significant difference at $p < 0.05$.

The expression profile of the *TP53* gene was significantly increased in the propofol group, when comparing before and after anesthesia ($p = 0.029$). For the other genes, there was no significant difference in the expression profile in any of the study groups (see Table 2).

**Table 2.** A comparison of gene expressions in study groups, where * is the *p*-value when comparing T1 with T0.

| Trial Groups / Gene Analyses | Sevoflurane * | Sevoflurane + Lidocaine * | Propofol * | Propofol + Lidocaine * |
|---|---|---|---|---|
| *TNF-α* | 0.877 | 0.253 | 0.367 | 0.241 |
| *VEGFA* | 0.549 | 0.291 | 0.515 | 0.641 |
| *CASP3* | 0.569 | 0.276 | 0.947 | 0.876 |
| *BCL2* | 0.311 | 0.436 | 0.711 | 0.915 |
| *SMAD4* | 0.874 | 0.976 | 0.396 | 0.345 |
| *MMP7* | 0.592 | 0.128 | 0.965 | 0.426 |
| *TP53* | 0.168 | 0.760 | **0.029** | 0.710 |
| *TGF-β1* | 0.529 | 0.334 | 0.466 | 0.574 |

As shown in Figure 5, for most of the gene expression levels, except TP53 in the propofol group, the results were not statistically significant.

### 3.6. Clinical Outcome

We did not intend to focus on clinical outcomes in this study. However, even if there were different lengths of postoperative time intervals since surgery, all patients included in the study reached one year after surgery without any clinical evidence of recurrence.

## 4. Discussion

Our study assessed the effect of two anesthetic techniques on different cell functions in the HCT 116 cancer cell in vitro by using serum from patients undergoing colorectal cancer surgery under propofol–TIVA or sevoflurane anesthesia with or without lidocaine.

Regarding cell viability, the only significant difference we noticed was found in the sevoflurane–lidocaine group, where cell viability significantly increased at T1 compared to T0 ($p = 0.028$). The viability levels also increased in the other study groups but did not reach statistical significance. Our results are similar to those reported by Jaura and Buggy in breast cancer cells where they found no significant difference in the effect of serum from patients with sevoflurane–opioid vs. propofol paravertebral blocks on breast cancer cell viability [16]. We do not have an explanation for any particular result in the sevoflurane–lidocaine group where we expected quite the opposite results according to our previous studies and other in vitro studies on lidocaine [19,27]. We did not reach an interpretation for the significant differences in the SL group, except a lower dose of lidocaine infusion, as compared with recommended infusion rates and plasma concentrations, respectively, which may not have had effects on the cell viability levels reported so far for lidocaine. Moreover, most of the in vitro studies used higher lidocaine doses than those regularly used in clinical practice.

Different results showing decreases in cell viability levels were reported by Tat et al. in an in vitro study which showed that both propofol and lidocaine had antiproliferative tumor effects, both dose- and time-dependent, without affecting the tumor microenvironment, on colon cancer cells (HCT-116 and RKO cells) [19]. These different results may be explained by the higher concentrations of propofol (2–4 µg/mL) and lidocaine (2–4 µM) used in the in vitro study rather than those clinically used in this study.

Similarly, different results to ours have been published by other in vitro studies, showing a reduction in the viability and proliferation of colon cancer cells (HT-29, SW480, and LoVo) with local anesthetics [28,29], but at higher local anesthetic concentrations than those clinically used. Thus, Bundscherer et al. did not find significant proliferation inhibition results of HT-29 and SW480 cells for concentrations of lidocaine of 1–1000 µM after 48 h of exposure (results that are similar to ours), but cell line proliferation was inhibited for lidocaine concentrations of 10 and 100 µM in SW480 [27]. Ropivacaine inhibited proliferation of HT-29 cells at concentrations of 250 and 550 µM [29]. Propofol increased the activity and tumor-killing ability of NK cells in patients with colon cancer [30] and, in association with local anesthetics, they were administered as epidural analgesia, inhibited proliferation and invasion, and increased apoptosis in LoVo colon cancer cells at targeted plasma concentrations of 3.5–4 µg·mL$^{-1}$ in the Marsh model [31].

Apoptosis is a form of programmed cell destruction separate from regular cell death. Our study aimed to investigate the effect of the anesthetic technique on the serum's ability to influence apoptosis using fluorescence microscopy apoptosis. Our results showed no statistical difference between post- and pre-operative apoptosis in each group and between study groups. Similar results to ours were reported by Bundscherer et al. using different concentrations of lidocaine exposure (10–1.000 µM) in two different colon cancer cell lines (HT-29 and SW480), where only decreased apoptosis was seen after 3–24 h of incubation in the SW480 cell line [28].

The results reported by Xu et al., who used LoVo colon cancer cells and compared sevoflurane–opioids with propofol–epidural anesthesia, were different from ours [31]. They analyzed serum from patients 24 h after operation, while epidural ropivacaine was administered in the propofol group. Under these conditions, their study reported significantly decreased proliferation and invasion levels and significantly increased apoptosis levels in the propofol–epidural group [31].

The effects of the anesthetic technique on apoptosis in our study were insignificant compared to those reported in other studies, which may be due to the short exposure time to anesthetic agents; thus, we used blood drawn at the end of anesthesia. The low infusion rate for lidocaine in certain cases may have also contributed to these results, as well as the type of colon cancer cells used in our study (HCT 116). For breast cancer, in an in vitro

study, Jaura et al. found that the apoptosis of the human breast cancer cell line MDA-MB-231 was reduced when cells were exposed to the postoperative serum of standard general sevoflurane–opioid anesthesia, as compared with the propofol–paravertebral block when venous blood was collected 1 h after surgery [16].

We further investigated the influence of two anesthetic techniques with or without lidocaine on cell cycle phases (G1, S, and G2), assessed with flow cytometry. We did not detect any significant difference in colon cancer cell migration and cell cycle phases when exposed to patient serum from all study groups. To the best of our knowledge, this is the first study to investigate the effects of TIVA and sevoflurane anesthesia with or without lidocaine infusion on cell cycle in HCT-116 colon cancer cells.

When assessing the effects of the anesthetic technique on colon cancer cell migration using the wound scratch assay, we did not find any difference between the postoperative and preoperative effects of serum from study groups as well as between study groups. Similar results were reported by Deegan et al.'s study on breast cancer cells (ER-MDA-MB-231), which showed that serum from patients subject to two different anesthetic techniques did not influence cell migration using the wound scratch assay [14]. However, in an in vitro study on breast cancer cells, D'Agostino et al. showed different results in the cell migration and scratch wound assay. In their study, they found that lidocaine, in clinical concentrations, could block through CXCR4 cell migration, causing a significant reduction in cancer cell motility and wound assay scratch [32].

Lastly, we assessed different gene expression levels of the most common implicated genes in colorectal cancer after exposure to two types of general anesthesia with or without lidocaine. To our knowledge, only one study approached multiple autophagy-related genes (ATGs) in a pilot study where it was found that there was a different expression of multiple genes in the autophagy pathway in both early- and late-stage colorectal cancer and where several genes related to apoptosis such as *BCL2* and *CASP3* were down-regulated [33]. In our study, *CASP 3* was increased in the SL group and decreased in the S group, while for the rest of the study groups, there were no significant differences. For the *BCL2* expression, there was no significant difference in the study groups, but expression was increased in lidocaine groups. In a recent study, Luo et al. showed that pro-apoptosis factor *CASP3* and anti-apoptosis factor *BCL2* are implicated in the early apoptosis stage and can maintain a balance following apoptotic development [34].

The *TP53* gene on human chromosome 17 with its p53 protein fraction is considered to have tumor suppression functions mediated throughout various forms rather than a single pathway or transcription. The inactivation of the *TP53* gene, which is said to occur following the negative regulation of wild type p53 proteins or by forming mutations, gives rise to invasion, proliferation, and cell survival, which will contribute to cancer progression and metastasis [35–37].

In our study, *TP53* expression was significantly increased in the propofol group, while variations in the expression of *TP53* were not statistically significant in the sevoflurane groups. Our group previously found that lidocaine significantly increased p53 in the cell cultures of colon cancer [19]. It is possible that the different results on lidocaine in this study may be attributable to lower plasma concentrations of lidocaine (the mean infusion rates during surgery were 1.62 mg/kg/h and 1.51 mg/kg/h, respectively) than those used in our previous study where the IC50 used for lidocaine in HCT-116 was 948.6 μM, thus showing compatibility with those used during infiltration.

Our study has some limitations. Our results may have been influenced by using only one colon carcinoma HCT116 cell line. It is possible that using different colon cancer cell lines—RKO and HCT 15—may have produced different results. Another limitation is the small size of study groups, although it is bigger than that used by most similar studies [16]. Another important limitation is that we did not measure the plasma concentration of propofol, but we found that the mean highest effect site concentration could be estimated as 2.99 mcg/mL (Table 1). The plasma concentration of lidocaine was not measured as well, and it could have been lower than the minimal effective plasma concentration (that

is generally 3–5 mcg/mL) [38] due to precautious measures being exercised by certain patients (bradycardia, hypotension, etc.), where we decreased the infusion rate (lower than 2 mg/kg/h) according to the study protocol, such that the mean infusion rate was 1.62mg/kg/h. Thus, the estimated plasma concentration of lidocaine may have been lower that 3 mcg/mL. In our previous study on the effects of lidocaine in HCT-116 colon cancer cells, the IC50 for lidocaine in HCT-116 was 948.6 μM, which was much higher than effective clinically used concentrations [19,39]. Another limitation of our study may have been the short exposure time to anesthetic agents, especially to lidocaine. The effect of anesthetic agents on cancer cells is reported in some studies as being both time- and dose-dependent [19,27]. We collected blood at the end of the operation and not after 24 h, unlike other studies [14,15]. However, there are studies that have used similar time intervals to us (1 h) in breast cancer [16], while others have used different time intervals [14,15]. It is possible that the effect of anesthetic agents may be dependent on the length of exposure and the tumor type. Confounding factors such as large interindividual variations in immune and inflammatory responses to cancer and surgery and different cancer stages, although similar between study groups, may have interfered with our results. The use of opioids (fentanyl and morphine) may have also interfered with immune responses to cancer as well, though they were similar in our study groups. Similarly, surgical techniques may also influence the stress response to surgery and this may be another source of confusion bias. We did not include any robotic surgery in our study and had a limited number of laparoscopies; however, these were similar in all study groups.

However, to our knowledge, this is the first study to investigate the effects of serum from colon cancer surgical patients undergoing propofol–TIVA vs. sevoflurane anesthesia with or without intravenous lidocaine on HCT-116 cancer cell biology and cancer gene expressions.

## 5. Conclusions

In conclusion, in our study, the serum from patients subject to different anesthetic techniques did not significantly influence the apoptosis, migration, and cell cycle of HCT-116 colorectal carcinoma cells. The increased expression of *TP53* genes in the propofol group is consistent with the results of similar in vitro studies, and may be one of the mechanisms by which anesthetic agents may influence cancer cell biology. Further studies that investigate the effects of propofol and lidocaine in different plasma concentrations on different colon cancer cell lines and the assess the impact of these findings on clinical outcomes are urgently needed.

**Author Contributions:** Conceptualization, A.L.A., C.T. and D.I.; formal analysis, A.L.A., A.J., C.T. and D.I.; funding acquisition, A.L.A.; investigation, A.L.A. and D.I.; methodology, A.L.A., A.J., C.T., R.-M.H. and D.I.; resources, A.L.A. and R.-M.H.; supervision, C.T. and D.I.; visualization, A.B.T.; writing—original draft, A.L.A. and A.J.; writing—review and editing, A.L.A., A.J., A.B.T. and D.I. All authors have read and agreed to the published version of the manuscript.

**Funding:** The study was financed by an internal grant from the University of Medicine and Pharmacy "Iuliu Haţieganu" Cluj-Napoca: doctoral research project no.1529/3/18 January 2019.

**Institutional Review Board Statement:** The study was conducted in accordance with the Declaration of Helsinki and approved by the Ethics Committee of the University of Medicine and Pharmacy "Iuliu Haţieganu" Cluj-Napoca (Nr. 436 from 26.11.2018).

**Informed Consent Statement:** Informed consent was obtained from all subjects involved in the study.

**Data Availability Statement:** Not applicable.

**Acknowledgments:** The authors would like to thank Iurie Acalovschi, Chair President of the Research Association in Anesthesia and Intensive Care (ACATI), Cluj-Napoca, Romania, for the publishing grant offered.

**Conflicts of Interest:** The authors declare no conflict of interest.

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
