# Peer review of "The Effect of Different Anesthetic Techniques on Proliferation, Apoptosis, and Gene Expression in Colon Cancer Cells: A Pilot In Vitro Study"

_cimb, doi:10.3390/cimb45010049_

Round 1

Reviewer 1 Report

Congrtulation and wish you Good luck

Author Response

We would like to thank you for your effort in reviewing our manuscript.

Reviewer 2 Report

I have few minor concerns about this manuscript.

1. Abbreviations should be spelled out, eg TIVA etc...

2. Catalogue number of all reagents should be provided.

3. Why Hoechst was used for apoptosis, I guess there are many better reagents available for such experiments.

4. All experiments were done in 24-hour time point, I think even for many drugs also do not show impact within this time period. Incubating for longer time like 72 hours or more may have some influence on observed data.

5. Data representation is highly poor need the replotting of all graphs with sufficient knowledge and self-explanatory.

6. This will be better if the authors could test the gene expression in surgically isolated tumors from patients.  What were the importance of testing these genes in this study? any particular postulated mechanism authors could explain?

Author Response

Thank you much for considering our manuscript. 

We  considered all suggestions and here are the corrections we made to our manuscript.

  1. Abbreviations should be spelled out, eg TIVA etc...

            Response. We spelled out most abbreviations where they appear for the first time in the text starting with the title, abstract and main document of the manuscript.

e.g., “Our main objective was to compare the effect of propofol-total intravenous anesthesia (TIVA) with sevoflurane anesthesia and to investigate the potential role of intravenous lidocaine on colon cancer cells functions.”

“A number of studies, mostly in vitro or retrospective clinical studies, compared the two main anesthetic techniques -total intravenous anesthesia (TIVA) and inhalation anesthesia- and found that TIVA is followed by a better long-term survival,”

  1. Catalogue number of all reagents should be provided.

            Response. Catalogue number of the reagents were introduced in the text and also in the supplemental material.

  1. Why Hoechst was used for apoptosis, I guess there are many better reagents available for such experiments.

            Response. “We used Hoechst solution as a stain for apoptotic cells due to its binding to chromatin and its fluorescent signal. Thus, using Hoechst we obtained images from a large area of the flask, allowing us to determine the percentage of the apoptotic and non-apoptotic cells using our fluorescent inverted microscope.”

 Sure, for apoptosis evaluation are many other reagents available on the market, however for our study we decided that Hoechst is suitable due to the high number of samples.

  1. All experiments were done in 24-hour time point, I think even for many drugs also do not show impact within this time period. Incubating for longer time like 72 hours or more may have some influence on observed data.

            Response.  Indeed, a longer incubation time may have changed the results but on the other hand, in clinical practice, during non-eventful perioperative period, it is only lidocaine that may be administered as a 72 hrs infusion, so, by extending exposure time we would not have been close to clinical practice.

Most of the studies on this topic (despite being a very few) used 24 hrs as incubation time, probably because of the same reasons. e.g., Jaura AI, Flood G, Gallagher HC, Buggy DJ. Differential effects of serum from patients administered distinct anaesthetic techniques on apoptosis in breast cancer cells in vitro: a pilot study. Br J Anaesth. 2014 Jul;113 Suppl 1:i63-7. doi: 10.1093/bja/aet581.  Sun C, Liu P, Pei L, Zhao M and Huang Y (2022) Propofol Inhibits Proliferation and Augments the Anti-Tumor Effect of Doxorubicin and Paclitaxel Partly Through Promoting Ferroptosis in Triple-Negative Breast Cancer Cells. Front. Oncol. 12:837974. 

  1. Data representation is highly poor need the replotting of all graphs with sufficient knowledge and self-explanatory.

            Response. Thank you very much for your interest in our data. Please be more specific regarding the figure changes you would prefer. Faceting a plot is a rather basic approach and should not pose a problem to your average reader.

  1. This will be better if the authors could test the gene expression in surgically isolated tumors from patients.  What were the importance of testing these genes in this study? any particular postulated mechanism authors could explain?

            Response. Thank you. Indeed, it would have been of interest such a study. However, we wanted in the beginning to test if patients’ serum would have different antiproliferative effects according to the type of anesthesia. As a preliminary study, we evaluated just a few of the genes involved in cell death, survival and based on these data we wish to further evaluate more biological pathways via the genes that encode key proteins in angiogenesis, apoptosis, immune response and then we can choose if the study can be shifted to gene expression in tumors from patient biopsy. The genes tested in our paper are important for understanding if and how sevoflurane, lidocaine, propofol and the presented combinations can induce cell death in tumor cells.

Reviewer 3 Report

You find attached my comments on the paper. 

Author Response

Thank you much for considering our manuscript.

We  considered all suggestions and here are the corrections we made to our manuscript.

Reviewer 3

  1. The title should definitively be shorten The title is very long (33 words, 198 characters without spaces, 230 characters with spaces). I think it should be shortened depending on the objectives of the paper. The wording in the title are also problematic “colorectal surgical cancer” “TIVA” as an abbreviation for “total intravenous anesthesia” – this abbreviation appears firstly in the tile and then in the Abstract (line 23) and in the Introduction (line 53) without being explained. This might be a widely used abbreviation in Anesthesia and a journal in this field may allow its use without explanation, but this is a journal of Molecular Biology where the audience background is very diverse. “TIVA or Sevoflurane anesthesia with or without intravenous lidocaine” - this is a shorthand for the anesthesia protocol. I suggest a more general formulation e.g. “various anesthesia schemes”

            Response. Thank you for this justified observation. We explained and spelled TIVA in all indicated places (see answer to reviewer 2). Title was shortened and is now 21 words.

  1. Using “despite” is not appropriate since the two statements in the phrase do not seem to be contrastive. “treatment for colorectal cancer remains surgery” – this is a highly problematic statement. Most of the time the treatment of colorectal cancer is multimodal involving surgery and adjuvant therapy in T3-T4 stages and in metastasis suspicions, for example. It is redundant to add that in rectal localization the treatment sometimes involves radiotherapy. “colorectal” – Is this this term appropriated in the present case given the fact that only 1 case included in this study is actually a rectal cancer case. Moreover, the NCCN for example distinguishes between colon and rectal cancer in terms of therapeutic recommendations. If one focuses on colon, then it should be specified that surgery is indeed the main course of treatment, but depending on staging adjuvant therapies are administered

            Response. Indeed we agree with this statement. We corrected accordingly in the abstract.

“Abstract: Background: Colorectal cancer has a high prevalence and mortality. Treatment for colorectal cancer is multidisciplinary but in most cases the main option remains surgery. Intriguingly, in the last years a number of studies showed that patient’s postoperative outcome may be influenced by certain anesthetic drugs.”

However, we would like to mention that most of the patients are operated and our study, as many at present, focus on the effects of anesthetic drugs on tumor progression and metastasis postoperatively. It is intriguing that even if the patient is exposed to anesthetic drugs a couple of hours, this may have an impact on postoperative outcome in terms of tumor progression. Moreover, there are a few studies showing that by adding lidocaine to chemotherapeutical drugs may decrease tumor progression.

  1. If something could influence the anesthesia protocol, it would rather be the post-operative evolution and possibly local or distant recurrences after a short period after surgery. Survival at 5 years is rather influenced by clinico-demographic and genetic factors. A sentence to that effect should be added. Anesthetic substances have a modulatory role in the tumor microenvironment, but they cannot be used as predictors for long-term evolution. This is admitted even by the authors.

            Response. Indeed, survival at 5 years may be influenced by a number of clinic-demographic and genetic factors, but the hypothesis that anesthetic drugs may influence survival and DFS is intriguing and this is the reason for which many studies focused on this in the last years. This may be due to reducing the number of circulating tumor cells or by influencing immune response perioperatively, thus this may fight and slow tumor progression.

We added the indicated sentence in the Introduction section.

“Recent data have shown that during perioperative period, patient’s immune system and cancer cells functions may be influenced by anesthetic drugs, some of which have shown to have an impact on cancer recurrence and metastasis [2,3,4], although long term survival may be influenced by numerous clinic-demographic and genetic factors.”

  1. The authors do not mention whether they used curarization to maintain anesthesia. If they used it, how did they monitor its administration?

            Response. This was mentioned in the text. We added now that additional boluses were administered when necessary and the fact that we did not monitor objectively neuromuscular block.

“All patients included in the study were administered a prophylactic dose of LMWH 12 hours prior to surgery. Anesthesia was induced in all groups with fentanyl 2-3 μg/kg, propofol 1.5-2 mg/kg and atracurium/rocuronium 0.5-0.6 mg/kg (additional boluses were given when necessary at anesthesiologist's discretion; curarisation was not objectively monitored)”

  1. This information is very important. It is essential to know how the antiproliferative effects of the serum were detected. Without the presence of these effects, valid testing of the working hypothesis cannot be demonstrated, i.e. the existence in the serum of patients of some factors directly related to the agents for maintaining anesthesia that would actually have an impact on the initiation by means of tumor cells circulating in the bloodstream of metastases. It would not be bad for the authors to insist with a sentence or two in the "Introduction" to describe how the literature demonstrates the direct antiproliferative effect of anesthetic agents or indirectly of some of their metabolites.

            Response. Thank you for the suggestion. We introduced the suggested sentence in Introduction.

“It has been shown by a number of studies that propofol has direct antiproliferative and apoptotic effects in different cancer cells as well as indirect effects by diminished or absent immunosuppressive effects when compared with inhalation agents [6].

  1. Although the statistical differences in SLT1 vs SLT0 cell viabilities may be correct, it is difficult to grasp why this might be true. I think a comparison of cell viabilities ST0 - PT0 - SLT0 - PLT0 (let's say by ANOVA test) should be performed to demonstrate that they represent similar or different starting points. Then ST1 - PT1 - SLT1 - PLT1 to show that different results are obtained between them. If ST0, PT0, SLT0 and PLT0 are not different with acceptable statistical significance then the pairwise comparison as shown in Figure 1 is legitimate. If the cell viabilities induced by ST0, PT0, SLT0 and PLT0 show differences then it is recommended to make a comparison of the ratios of ST1/ST0 - PT1/PT0 - SLT1/SLT0 - PLT1/PLT0 viabilities.

There still remains the problem of objectifying the presence of lidocaine or any anesthetic or metabolite thereof in the serum of patients. I think this major flaw makes the presented results less credible. The methodology presented above can, however, partially compensate for the absence of such clarifications.

Even if the authors find a way to fix this flaw, I think that the time T1 (at the end of surgery) is not very well chosen: a good part of the anesthetic agents will have been metabolized at the end of surgery. A more intelligent strategy would have been to sample serum at induction and maintenance and at the completion of surgery. Then it would be determined which of the series of samples have higher concentrations of lidocaine or another metabolite with an anti-proliferative role. Another (obviously more expensive) route would have been to determine the cell viability induced by the sera sampled at the above time points.

            Response. Thank you for the suggestion. Statistical analysis was done with ordinary ONE WAY ANOVA with Turkey’s multiple comparison test . The evaluation of T0 and T1 groups independent, by applying Ordinary One Way ANOVA multiple comparison with Turkey’s test, no statistical significance was observed in T0 groups, while in T1 groups, we observed a significant difference of P= 0.0022 between LOT A and B, and P=0.0002 between LOT B and C, and P=0.0010 between LOT B and D. Data representation as figures will be added in the resubmitted supplemental material.

Regarding the plasma level of anesthetics, it is true that we did not measured plasma concentration of anesthetic agents in this study and we mentioned this in Limitations. But propofol mean effect site concentration was 2.88 and 2.58 µg/ml respectively with no statistical difference between groups. The duration of operation allowed pharmacological equilibrium between blood and ce of propofol set by anesthesiologist. The plasma concentration of propofol at the end of operation is, in our view not so important, in terms of antiproliferative effects and none of the studies focused on this topic measured this. In fact, in the literature none of the studies looked at plasma concentration of propofol, but all evaluated recurrence rate after TIVA when compared with inhalation in general. There was no difference in lidocaine mean rate of infusion in study groups, while the duration of surgery allowed reaching pharmacological equilibrium between compartments.

On the other side you may be right. We published one of our study in which we showed that antiproliferative effects of propofol and lidocaine on colon cancer cells were time and dose dependent but at doses far from those clinically used. So you are right, probably in the future, once the beneficial effect of TIVA and lidocaine will be demonstrated, studies will have to focus on length of time and minimal "effective" dose to have such antitumor effects. If we would have sampled patients during surgery it would have added more confounding factors like the right operatory time, duration since the beginning of surgery, etc; such a study, on the other side, would have been very expensive, financially. Actually, we tested at the beginning each sample's antiproliferative effects and the differences were huge between patients. This is why we, as all the other authors (e.g., Donal Buggy and his team), tested pooled serum at the beginning and the end of surgery with all the confounding factors that may have affected the results.

  1. The Figure 1 caption is not very lisible. I would suggest the change

The study groups are as follows: TIVA with lidocaine (PLT0 - before surgery and PLT1 after surgery); TTVA (PT0 - before surgery and PT1 after surgery); Sevoflurane with lidocaine (SLT0 - before surgery and SLT1 after surgery); Sevoflurane (ST0 - before surgery and ST1 after surgery).

An even better idea would be to enter these labels/abbreviations on line 116 as well either as a text, or as a small table

            Response. we do have this text at the beginning of Methods: Patients were randomized with the help of a computer-generated number table into four study groups of 15 patients each, as follows: group S – sevoflurane, group P – TIVA, group PL – TIVA with lidocaine, group SL – sevoflurane with lidocaine. However we repeated group allocation where suggested.

“Results on cellular viability in study groups, S – sevoflurane, P – TIVA, PL – TIVA with lidocaine, and SL – sevoflurane with lidocaine, are shown in Fig. 1.”

  1. These are the p-values. Table 2 caption should be modified to explicitly indicate this. Authors should add a reference to Figure 5.

            Response. Thank you for this observation. We added this correction at Table 2.

“Table 2. Comparison of gene expression in study groups, where * is p-value when comparing T1 with T0.”

  1. Authors should specify that the representation is in logarithmic scale

 Response. We mentioned this in Figure 5 caption.

  1. I do not understand why the authors want to add these clarifications considering that the differences (positive or negative) are not statistically significant.

Response. We corrected this by changing with the sentence: "As shown in Fig. 5, for most of gene expression levels, except TP53 in propofol group, the results were not statistically significant.”

  1. With this statement, the authors themselves acknowledge that a more rigorous analysis of anesthetic levels, in the serum of patients is needed.

Response. Indeed, determining plasma concentration of lidocaine would have been of great help. However, since this was a pilot study and we expected some results without measuring plasma concentration. For the future we definitely plan to measure plasma concentration of lidocaine at least in some patients, because financially, such a study is very expensive. For propofol plasma concentration may be approximated by the targeted ce/cp of propofol. Besides it is only lidocaine that is recommended for an extended i.v. infusion postoperatively, while propofol is stopped at the end of operation. We added a few more comments in Discussion section on this aspect.

“We have not reached an interpretation for the significant difference in SL group, except a lower dose of lidocaine infusion as compared with recommended infusion rates and plasma concentration, respectively, that may not have had the effects on cell viability reported so far for lidocaine. Moreover, most of the in vitro studies used higher lidocaine doses as those used regularly in clinical practice.”

“Another important limitation is that we did not measure plasma concentration of propofol,”

  1. This similarity between the results obtained by exposure to the serum of patients undergoing different types of anesthesia collected at T0 and T1 and the results obtained by exposure of cell cultures to precise concentrations of lidocaine needs further explanation. Mere observation of this similarity is not enough.

            Response. Thank you for this very justified suggestion. We added the required information in this paragraph.

“Similarly, different results as ours have been published by other in vitro studies showing a reduction in viability and proliferation on colon cancer cells (HT-29, SW480, LoVo) with local anesthetics [28,29] but at higher concentrations of local anesthetics than those clinically used. Thus, Bundscherer et al did not find significant proliferation inhibition of HT-29 and SW480 cells for concentrations of lidocaine of 1-1000 µM after 48 hours of exposure (results that is similar with ours), but in SW480 cell line proliferation was inhibited for lidocaine concentration of 10 and 100 µM [28]. Ropivacaine inhibited proliferation of HT-29 cells at concentrations of 250 and 550 µM [29]. Propofol increased activity and tumor-killing ability of NK cells in patients with colon cancer [30] and in association with local anesthetics, administered as epidural analgesia, inhibited proliferation and invasion and increased apoptosis in LoVo colon cancer cells at targeted plasma concentration of 3.5-4 μg.ml−1 in Marsh model [31].”

  1. The conclusions of the paper are just a small summary of the results. I think the authors should re-emphasize one of the main limitations, which is that they did not measure serum anesthetic levels.

  1. Also, given that this study continues a series of studies on serum-exposed cell lines from surgical oncology patients that concluded no statistically significant differences, it would be preferable if the authors expressed themselves in a sentence or two with regarding the usefulness of this line of investigation and/or changes in the investigation methodology.

Personally, after reading the references provided by the authors in the reference list, I cannot declare myself convinced and satisfied with this line of investigation.

Response. We changed Conclusions according to the suggested correction. Thank you for suggestion. We also added a sentence on the originality of our study after Limitation section.

“In conclusion, in our study, serum from patients receiving different anesthetic techniques did not influence significantly apoptosis, migration and cell cycle of HCT-116 colorectal carcinoma cells. Increased expression of TP53 gene in propofol group is consistent with the results of similar in vitro studies and may be one of the mechanisms by which anesthetic agents may influence cancer cells biology. Further studies to investigate the effects of propofol and lidocaine in different plasma concentrations on different colon cancer cell lines and the impact of these findings on the clinical outcome are much needed.”

In the end we would like to express our point of view regarding our investigation and the way we correlate this with the literature.

In the last years a number of studies investigated if anesthetic agents may influence cancer biology and postoperative outcome in cancer patients undergoing surgery. breast cancer was mostly approached and for this cancer it is not clear that anesthetic technique does not influence outcome when comparing propofol- TIVA with inhalation anesthesia. For the other types of cancers studies are ongoing, in terms of comparing TIVA with inhalation. The most cited (probably) study is Wigmore's on 7000 patients who demonstrated retrospectively that TIVA ameliorated outcome and mortality. apart from comparing TIVA with inhalation, lidocaine and other local anesthetics are now investigated for their potential impact on cancer cell biology.

We also approached this topic on a number of studies published in JGLD (hepatocarcinoma) and JBUON (on colon cancer cells).

Out present study is a continuation of the former study on colon cancer cells, and in this study we moved toward clinical studies and we used serum from colon cancer patients undergoing surgery. It is true that the results of this pilot study were not so significant like for in vitro studies but give us an idea how to proceed further in our investigation, including some ideas you gave us. An argument for the need to continue our study is given by some recent studies on this topic:

1. Wang, Z., Cao, B., Ji, P., and Yao, F. (2021). Propofol inhibits tumor angiogenesis through targeting VEGF/VEGFR and mTOR/eIF4E signaling. Biochem. Biophys. Res. Commun. 555, 13–18. doi:10.1016/j.bbrc.2021.03.094

2. Qu, L. Yang, Q. Shi, X. Wang, D. Wang, G. Wu. Lidocaine inhibits proliferation and induces apoptosis in colorectal cancer cells by upregulating mir-520a-3p and targeting EGFR. Pathol. Res. Pract, 214 (12) (2018), pp. 1974-1979

            The strongest point of this study is that we used serum from patients in 4 study groups 2 groups with TIVA and 2 with inhalation anesthesia, both with and without lidocaine and this is the first study with these 4 groups. We have a clinical ongoing study with the same groups (protocol published in Trials) and there is a large multicentre international study with the same 4 study groups in colon and lung cancer. The second strong point is the result on TP53 gene variation in propofol group confirming other results in vitro from the literature in which p53 was identified as a possible mechanism by which anesthetic agents may influence cancer cell biology.

Round 2

Reviewer 2 Report

1. Overall figures quality should be presented. I can see the X and Y-axis and labelling of graphs is not very clear and require a high-resolution figures image. I guess there are not too many figures in this manuscript and therefore authors can think of improving the quality of their work. Authors can think of changing the font-size, color of labelling in figures. Example figure 5 gene names are not very clear. 

2. Missing experimental replicates in figure 4 at least. No STD bar etc. 

Author Response

We would like to sincerely thank you for your effort and time in reviewing and improving our manuscript. 

  1. Overall figures quality should be presented. I can see the X and Y-axis and labelling of graphs is not very clear and require a high-resolution figures image. I guess there are not too many figures in this manuscript and therefore authors can think of improving the quality of their work. Authors can think of changing the font-size, color of labelling in figures. Example figure 5 gene names are not very clear. 

Response: Thank you for the remark and goodwill. We modified the quality of all figures ;they are now at 300 dpi. For figure 4 we introduced error bars and rebuilt the graphic and for fig. 5 we modified the font-size. We hope that the figures are now improved in quality.

  1. Missing experimental replicates in figure 4 at least. No STD bar etc. 

Response: We modified the legend of figure 4 :

“Cells migration ability was assessed on wound healing assay on HCT-116 cell line. Quantitative representation of the wound distance measurement in µm related to the time of treatment action. Data is represented as mean +/- standard deviation (n=3). Study groups: TIVA with lidocaine (group PL, before surgery-T0 and after surgery-T1); TIVA (group P, before surgery-T0 and after surgery-T1); Sevoflurane with lidocaine (group SL, before surgery-T0 and after surgery-T1); Sevoflurane (group S, before surgery-T0 and after surgery-T1).”

Reviewer 3 Report

Thank your for answering point-by-point to the questions I raised. 

A final point would be to correct the list of authors. Some of the authors are listed correctly: name(s) followed by surname(s), while other are listed with their surname(s), firstly and then with their name. 

Looking forward to see more of your research. 

Author Response

We would like to sincerely thank you for your effort and time in reviewing and improving our manuscript.  We have listed the authors in the correct order as the journal required. “Alexandru Leonard Alexa1,2,8*, Ancuta Jurj3, Ciprian Tomuleasa4,5,6, Adrian Bogdan Tigu 4, Raluca-Miorita Hategan2, Daniela Ionescu1,2,7,8

Round 3

Reviewer 2 Report

Dear Editor

Authors have addressed my concerns. No more questions from my end.